# Transcriptional regulation of Sis1 promotes fitness but not feedback in the heat shock response

Rania Garde[1,2], Abhyudai Singh[3,4,5,6]*, Asif Ali[1], David Pincus[1,7]*

[1]Department of Molecular Genetics and Cell Biology, University of Chicago, Chicago, United States; [2]Committee on Genetics, Genomics, and Systems Biology, University of Chicago, Chicago, United States; [3]Department of Electrical and Computer Engineering, University of Delaware, Newark, United States; [4]Department of Biomedical Engineering, University of Delaware, Newark, United States; [5]Department of Mathematical Sciences, University of Delaware, Newark, United States; [6]Center for Bioinformatics and Computational Biology, University of Delaware, Newark, United States; [7]Center for Physics of Evolving Systems, University of Chicago, Chicago, United States

**Abstract** The heat shock response (HSR) controls expression of molecular chaperones to maintain protein homeostasis. Previously, we proposed a feedback loop model of the HSR in which heat-denatured proteins sequester the chaperone Hsp70 to activate the HSR, and subsequent induction of Hsp70 deactivates the HSR (Krakowiak et al., 2018; Zheng et al., 2016). However, recent work has implicated newly synthesized proteins (NSPs) – rather than unfolded mature proteins – and the Hsp70 co-chaperone Sis1 in HSR regulation, yet their contributions to HSR dynamics have not been determined. Here, we generate a new mathematical model that incorporates NSPs and Sis1 into the HSR activation mechanism, and we perform genetic decoupling and pulse-labeling experiments to demonstrate that Sis1 induction is dispensable for HSR deactivation. Rather than providing negative feedback to the HSR, transcriptional regulation of Sis1 by Hsf1 promotes fitness by coordinating stress granules and carbon metabolism. These results support an overall model in which NSPs signal the HSR by sequestering Sis1 and Hsp70, while induction of Hsp70 – but not Sis1 – attenuates the response.

## Editor's evaluation

Here the authors describe an updated theoretical model describing how the Hsf1 transcription factor is activated in yeast in response to heat shock, and then test the new model experimentally, providing solid evidence that heat shock results in delayed folding of newly synthesized proteins (NSPs), which sequester the Hsp70 chaperone away from the inactive Hsp70/Hsf1 complex, releasing active Hsf1. They also demonstrate convincingly that the Hsp70 co-chaperone Sis1 is not required directly for the heat shock response (HSR), but under basal conditions targets Hsf1 to Hsp70 for repression and then upon upregulation by Hsf1 serves to promote fitness by coordinating stress granule and carbon metabolism to improve fitness in the presence of nonfermentable carbon sources. By taking into account NSPs and incorporating new roles for the Sis1 co-chaperone in their updated model, these studies represent a significant advance for the heat shock response field.

*For correspondence:
absingh@udel.edu (AS);
pincus@uchicago.edu (DP)

Competing interest: The authors declare that no competing interests exist.

## Introduction

Under steady-state growth conditions, cells have sufficient protein folding and degradation machinery to maintain protein homeostasis (proteostasis) (*Jayaraj et al., 2020*). However, in response to environmental stressors like a rise in temperature or exposure to oxidizing agents, the cellular demand for proteostasis machinery outstrips the supply, and cells activate the heat shock response (HSR). The HSR is a highly conserved gene expression program under the control of the transcription factor Hsf1 that coordinately increases expression of molecular chaperones and other proteostasis factors (*Pincus, 2020*). Implicated in both cancer and neurodegenerative diseases, modulation of the HSR could have powerful therapeutic benefits. Poor prognosis in human cancers – in particular breast and colon cancer – is correlated with hyperactive Hsf1 (*Puustinen and Sistonen, 2020*; *Wang et al., 2020*). At the other extreme, Hsf1 remains inactive in neurons derived from patients with neurodegenerative disorders despite the presence of protein inclusions and aggregates (*Chen et al., 2016*; *Gomez-Pastor et al., 2017*; *Jayaraj et al., 2020*). It is unclear how Hsf1 is mis-regulated in these disease states, and even the basic regulatory mechanisms that determine the activation dynamics of the HSR in healthy cells remain incompletely understood.

Due to its high degree of evolutionary conservation, many of the advances in elucidating the HSR and its regulatory mechanisms have come from studying the pathway in budding yeast. In yeast, the HSR comprises 42 genes activated by Hsf1 during environmental stress via binding to *cis*-acting heat shock elements (HSEs) in the promoter regions of these genes (*Pincus et al., 2018*). The Hsf1-induced genes encode a small but diverse set of proteins including chaperones localized to the cytosol, nucleus, endoplasmic reticulum, and mitochondria, translation regulators, and other protein quality control factors. In the absence of stress, Hsf1 is repressed by binding to the chaperone Hsp70, and dissociation of Hsp70 during stress releases Hsf1 to activate the HSR (*Kmiecik et al., 2020*; *Krakowiak et al., 2018*; *Masser et al., 2019*; *Peffer et al., 2019*; *Zheng et al., 2016*). Hsp70 binds to two distinct sites on yeast Hsf1 to mediate the repression (*Krakowiak et al., 2018*; *Peffer et al., 2019*). Biochemically, Hsp70 can dissociate both yeast and human Hsf1 from DNA (*Kmiecik et al., 2020*; *Masser et al., 2019*). However, Hsp70 may repress Hsf1 activity via other mechanisms as well, such as inhibiting interactions with the transcriptional machinery and disassembling transcriptional condensates (*Chowdhary et al., 2022*; *Zhang et al., 2022*).

While Hsp70 can bind to Hsf1 in the absence of any co-chaperones (*Krakowiak et al., 2018*; *Zheng et al., 2016*), the J-domain protein (JDP) co-chaperone Sis1 targets Hsf1 to Hsp70 in the nucleus to promote high-affinity binding, and nuclear localization of Sis1 is required to repress Hsf1 in the absence of stress (*Feder et al., 2021*). Upon heat shock, Sis1 re-localizes from the nucleoplasm to the periphery of the nucleolus, which may release Hsf1 from Hsp70-mediated repression and initiate the HSR (*Ali et al., 2022*; *Feder et al., 2021*). During prolonged stress, induction of the HSR feeds back to once again repress Hsf1, and this negative feedback specifically requires induction of Hsp70 (*Krakowiak et al., 2018*). However, it is unknown whether induction of any other HSR targets, such as Sis1, is also required for negative feedback.

Multiple studies have now implicated dissociation of Hsp70 from Hsf1 as the key regulatory event that activates the HSR, yet the clients that titrate Hsp70 away from Hsf1 following heat shock – that is the 'ligands' of the HSR – remain unidentified. Long assumed to be heat-denatured mature proteins (*Lindquist, 1986*; *Morimoto, 2008*; *Zheng et al., 2016*), it is notable that not a single endogenous eukaryotic protein has been shown to unfold following heat shock at the sublethal temperatures that elicit the HSR (*Yoo et al., 2022*). In fact, the protein aggregates that do form following physiological heat shock have been shown to largely consist of reversible biomolecular condensates that play an adaptive role in the stress response rather than proteotoxic aggregates (*Iserman et al., 2020*; *Riback et al., 2017*; *Wallace et al., 2015*). Moreover, several studies have demonstrated that depletion of amino acids or treatment with cycloheximide (CHX) substantially diminishes the output of the HSR following heat shock, suggesting that ongoing translation and/or newly synthesized proteins (NSPs) may drive HSR activation (*Masser et al., 2019*; *Triandafillou et al., 2020*; *Tye and Churchman, 2021*). Thus, while the identities of the HSR ligands have not been established, NSPs are currently thought to be a major category of HSR activators.

In this study, we incorporate these recent developments into an updated mathematical model of the HSR. First, we develop a new input function for the HSR that explicitly includes protein synthesis and maturation. In this input function, the effect of heat shock is to reduce the protein maturation rate,

resulting in an accumulation of NSPs. The NSPs in turn titrate Hsp70, resulting in the release of Hsf1 and activation of the HSR. This new input function allows the model to recapitulate the diminished output of the HSR following heat shock under conditions with reduced protein synthesis. Second, by incorporating the role of Sis1 in targeting Hsf1 to Hsp70 for repression under basal conditions, we were able to model the induction of the HSR upon depletion of Sis1 in the absence of stress. However, the model predicted that Sis1 does not function as a negative feedback regulator of the HSR. Indeed, genetic decoupling of Sis1 expression from Hsf1 regulation had no effect on HSR dynamics, validating the model prediction. Consistent with its lack of participation in deactivating Hsf1, pulse-label imaging of heat shock-induced Sis1 revealed that Sis1 is spatially localized away from Hsf1, while newly induced Hsp70 partially co-localizes with Hsf1. Finally, we show that while Sis1 is not involved in feedback regulation of the HSR, its Hsf1-dependent transcriptional regulation is required for fitness in the presence of nonfermentable carbon sources, and we link this growth phenotype to mis-regulation of stress granules. The new model we present for the HSR incorporates a revised, data-supported activation mechanism while providing further support for the core regulatory circuitry and concise feedback architecture we previously proposed. This work demonstrates that deactivation and basal repression of a pathway need not be achieved by the same mechanism, and that stress response target genes can differentially promote feedback and fitness.

## Results

### Modeling the role of ongoing protein synthesis in activation of the HSR

Previously, we proposed a model of the HSR based on a two-component feedback loop consisting of the chaperone Hsp70 and the transcription factor Hsf1 (*Krakowiak et al., 2018*; *Zheng et al., 2016*). In the model, Hsp70 binds and represses Hsf1 under basal conditions. Upon heat shock, unfolded proteins (UPs) accumulate and titrate Hsp70 from Hsf1, leaving Hsf1 free to induce expression of more Hsp70. Once Hsp70 has bound all the UPs, the excess Hsp70 could then bind to Hsf1 to deactivate the response. This model is capable of recapitulating HSR dynamics following heat shock, but it includes assumptions that are no longer consistent with the current literature.

The input for the original model was heat shock-generated UPs, with the underlying assumption that heat shock causes mature proteins to denature into UPs. However, it is no longer accepted that UPs activate the HSR. Rather, immature NSPs have been suggested to drive HSR activation (*Figure 1A*; *Feder et al., 2021*; *Gonçalves et al., 2020*; *Masser et al., 2019*; *Masser et al., 2020*; *Tye and Churchman, 2021*; *Tye et al., 2019*). Supporting this, Hsf1 target gene transcripts were depressed in pH-balanced, CHX-treated cells following heat shock relative to cells heat-shocked while actively translating (*Triandafillou et al., 2020*). Our analysis of these published RNA-seq data indicate that ongoing translation is required for more than 70% of the induction of HSR genes on average (*Figure 1B*). This suggests that NSPs – rather than denatured mature proteins – are major upstream activators of the response.

To account for translation dependent HSR activation in the model, we have implemented an input function in which there is a constant source of NSPs (i.e., a rate of ongoing translation) instead of a single initial input of UPs (*Figure 1A*, see methods). NSPs must then undergo a maturation step to become folded. In the updated model, heat shock inhibits the spontaneous folding rate of NSPs, rather than causing denaturation of folded proteins, thereby resulting in an accumulation of immature proteins. During heat shock, NSPs must interact with Hsp70 to either fold or be degraded. Heat shock still activates the HSR by titrating Hsp70 away from Hsf1, but by immature NSPs rather than by UPs.

Using this new input function, we tested the ability of the model to account for the effect of reducing translation prior to heat shock. To do this, we pretreated cells with rapamycin, which specifically inhibits expression of ribosomal proteins and biogenesis factors by inhibiting the target of rapamycin complex 1 (TORC1) (*Loewith and Hall, 2011*; *Figure 1C*). Together, these genes are estimated to constitute more than 40% of total NSPs in yeast under logarithmic growth conditions (*Ingolia et al., 2009*). The model predicts that such a reduction in NSPs should result in a concomitant reduction in HSR output following heat shock (*Figure 1D*). Indeed, using a yellow fluorescent protein (YFP) reporter of the HSR (HSE-YFP) (*Supplementary file 1*), we found that rapamycin-treated cells displayed a substantial reduction in HSR output across the heat shock time course (*Figure 1E*). To verify that the reduction in YFP fluorescence reflects a reduction in transcriptional activation of the

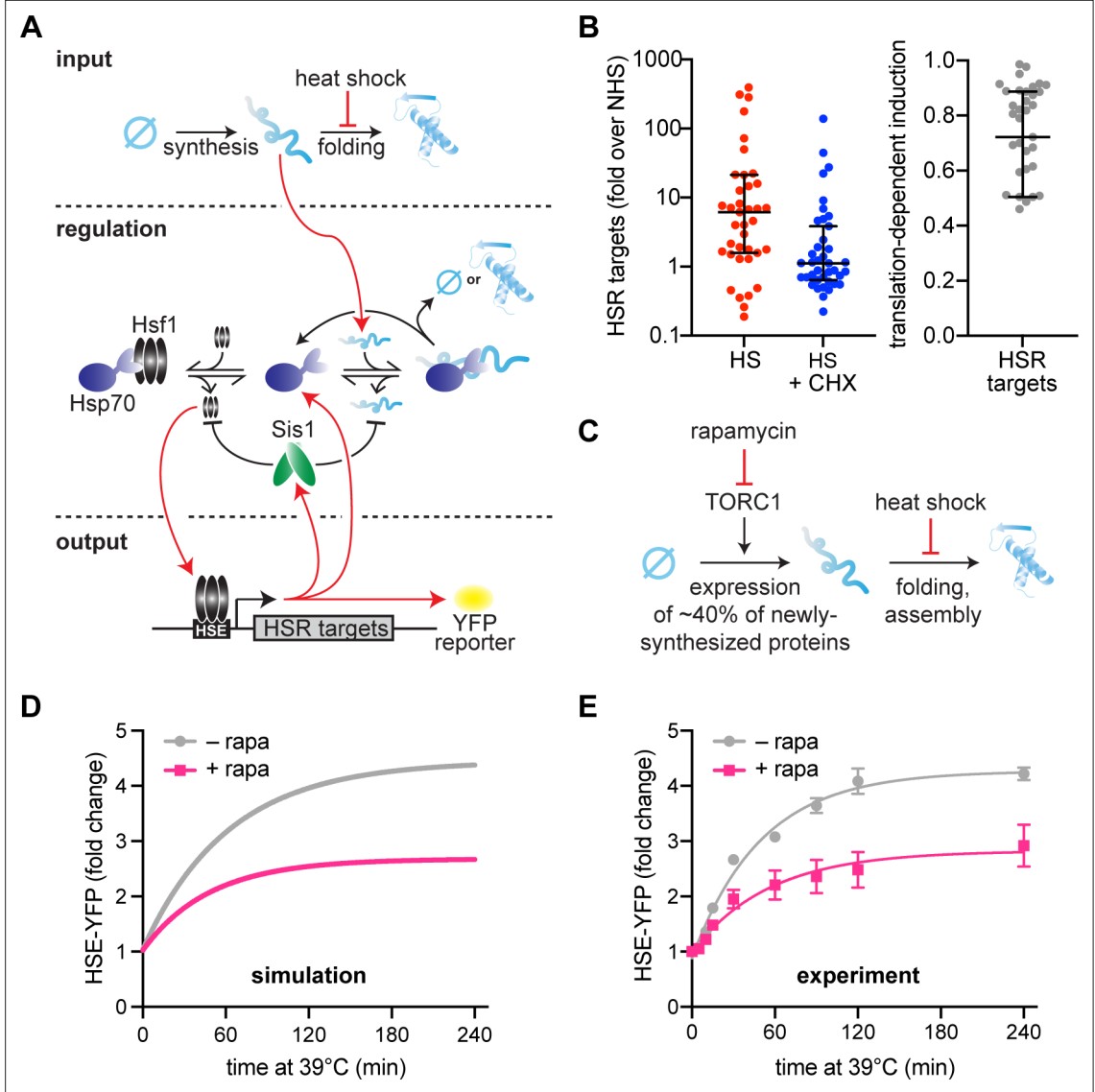

**Figure 1.** Newly synthesized proteins drive full activation of the heat shock response (HSR). (**A**) Schematic of the mathematical model. Heat shock stalls folding of newly synthesized proteins so newly synthesized unfolded proteins accumulate. Hsp70 is recruited to unfolded proteins (red arrow) in turn releasing Hsf1 to induce its transcriptional targets. Among these targets are Hsp70 and Sis1 (and HSE-YFP, our ectopic reporter of Hsf1 activity). Of note, Sis1 has two known roles. It substantiates Hsf1–Hsp70 binding and substantiates Hsp70-misfolded protein interactions. (**B**) Hsf1 target gene transcript levels, relative to non-heat shock levels. Cells were treated with 200 μg/ml cycloheximide (CHX; blue) and subjected to 42°C heat shock for 20 min. Untreated control (red) plotted for comparison (red). Raw data from *Triandafillou et al., 2020*. One minus the transcript levels of CHX-treated over untreated cells is the portion of transcriptional induction which depends on ongoing translation (right graph). Each data point represents an Hsf1 target gene. (**C**) Model of rapamycin effect. Heat shock stalls newly synthesized protein folding, and newly synthesized unfolded proteins trigger the HSR. Upstream, rapamycin inhibits target of rapamycin complex 1 (TORC1) which inhibits ribosomal protein production, causing a 40% decrease in newly synthesized proteins. (**D**) HSE-YFP levels during heat shock. Cells are pretreated with 10 μg/ml rapamycin 5 min before heat shock (pink) and compared to untreated control (gray). Each time point represents the average of the three biological replicates. Error bar represents the standard deviation of three replicates. (**E**) Simulation of HSE-YFP levels during heat shock with or without rapamycin.

The online version of this article includes the following figure supplement(s) for figure 1:

**Figure supplement 1.** Rapamycin reduces HSE-YFP at the mRNA level RNA levels during heat shock relative to non-heat shock, measured by qRT-PCR.

HSR (and not due to a reduction in translation of the YFP mRNA caused by rapamycin), we quantified YFP mRNA levels using qRT-PCR. Consistent with the fluorescence measurements, the mRNA was comparably reduced, indicating that the YFP levels accurately report on HSR activation in the presence of rapamycin (*Figure 1—figure supplement 1*). Together, these results demonstrate that the

updated mathematical model can capture the dynamics of the HSR during heat shock and recapitulate the effects of reduced protein synthesis on HSR output.

## Modeling the role of Sis1 as a basal repressor of the HSR

Recently, we found that Sis1, a JDP and Hsp70 cofactor, participates in repressing Hsf1 activity under non-heat shock conditions by promoting Hsf1–Hsp70 association (*Feder et al., 2021*). Consistent with this result, knockdown of Sis1 resulted in the highest level of activation of the HSR under basal conditions in a genome-wide CRISPRi screen (*Alford et al., 2021*). Moreover, it was independently shown that Hsf1 binds to Hsp70 as a typical client, via the Hsp70 substrate-binding domain in a nucleotide-regulated manner (*Masser et al., 2019*), further supporting the involvement of a JDP-like Sis1. Sis1 was also shown to potentiate the HSR in the presence of exogenous protein aggregates (*Klaips et al., 2020*). We incorporated Sis1 into the core circuitry of the model by representing the canonical role of JDPs in substantiating Hsp70-client binding: the Hsp70-client protein dissociation rate is inversely dependent on Sis1 concentration (see methods).

To assess this new model circuitry (*Figure 2A*), we first determined whether it could recapitulate the known role of Sis1 in preventing activation of the HSR under non-stress conditions. Experimentally, we transiently depleted Sis1 from the nucleus using the anchor-away system (*Feder et al., 2021*), and monitored Hsf1 activity via the HSE-YFP reporter. Upon nuclear depletion of Sis1, Hsf1 activity increased 2.5-fold compared to untreated control within 90 min (*Figure 2B*). Computationally, we simulated the model following instantaneous depletion of Sis1, which induced the HSR without a temperature upshift in agreement with the experimental result (*Figure 2C*). Notably, at later time points, the model failed to capture the sustained increase in HSE-YFP reporter we observed experimentally. This may indicate that prolonged depletion of Sis1 results in a breakdown in proteostasis that sustains HSR activity.

## Modeling experimental variability and HSR dynamics across a range of conditions

We implemented this updated mathematical model that incorporates the NSP input function and Sis1-containing circuitry to make predictions about the dynamics of the HSR. First, we repeated our standard heat shock time course – shifting cells from 30 to 39°C. We performed this experiment 18 times to train the model to include the range of experimental variation on different days (*Figure 2D*). Across these replicates, we observed variation on the order of 20% from experiment to experiment. Replicates performed on the same day behaved more similarly, so we hypothesized that a likely source of this variation could be the initial metabolic state of the cells on different days, reflecting differences in the amount of time they had been in stationary phase or outgrowth prior to initializing the heat shock. These metabolic differences would then be reflected in different rates of protein synthesis. Indeed, varying the rates of protein synthesis in the model yielded heat shock time course simulations that recapitulated the variation observed across the biological replicates (*Figure 2E*). Thus, by modulating a single parameter, the model was able to recapitulate both the average behavior and the experimental variability.

We next performed heat shock time course experiments and simulations at multiple temperatures ranging from 35 to 41°C. When we performed experiments across this temperature range, we observed a rapid induction phase at all temperatures, followed by a slow rise to a temperature-dependent maximal response (*Figure 2F*). Without changing any parameters, the model predicted that cells would reach different steady-state levels of HSE-YFP fluorescence in proportion to the increase in temperature, matching the experimental results (*Figure 2G*). Thus, the HSR is a 'dose-dependent' response, tuned quantitatively by the temperature of heat shock. Together, these results indicate that the mathematical model recapitulates experimental variation and the dynamics of the HSR across a range of conditions.

## Decoupling Sis1 expression from Hsf1 transcriptional activity

Previously, we found that transcriptional induction of Hsp70 is required to deactivate Hsf1 over a sustained heat shock time course, and we therefore deemed Hsp70 a negative feedback regulator (*Krakowiak et al., 2018*). Given that Sis1 promotes Hsp70-mediated repression of Hsf1 under basal conditions, we next tested whether induction of Sis1 is likewise required for Hsf1 deactivation

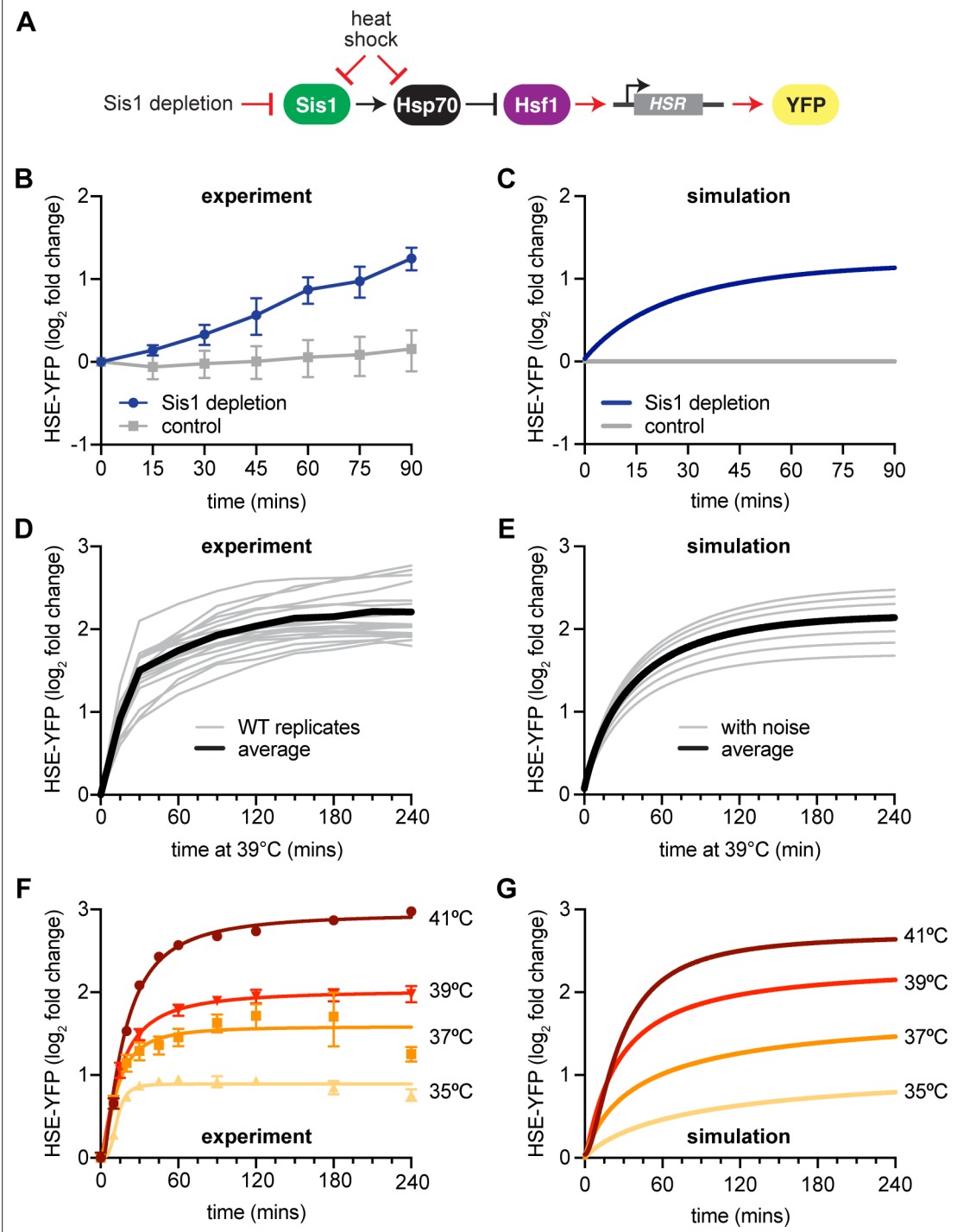

**Figure 2.** New heat shock response (HSR) model recapitulates experimental results. (**A**) Model of Sis1 effect. In non-heat shock, Sis1 promotes Hsp70 binding to Hsf1, repressing transcriptional activation of the HSR. Heat shock results in the titration of Hsp70 and Sis1, leaving Hsf1 free to induce the HSR and the ectopic reporter of Hsf1 activity, HSE-YFP (red arrows). (**B**) HSE-YFP levels after nuclear Sis1 depletion (blue) at 30°C, measured by flow cytometry. Sis1 was anchored away to cytosolic ribosomes using 10 μg/ml rapamycin and the time course was started immediately. (**C**) Simulation of HSE-YFP reporter after Sis1 depletion. (**D**) HSE-YFP expression in wild-type cells across 18 biological replicates (gray). Average in black. (**E**) Simulation of variation in the wild-type HSR due to metabolic differences leading to changes in basal translation rate. (**F**) HSE-YFP heat shock time courses at a range of heat shock temperatures between 35 and 41°C. (**G**) Simulation of heat shock time courses at different induction temperatures. Heat shock at higher temperatures was simulated by a proportional decrease in protein folding rate.

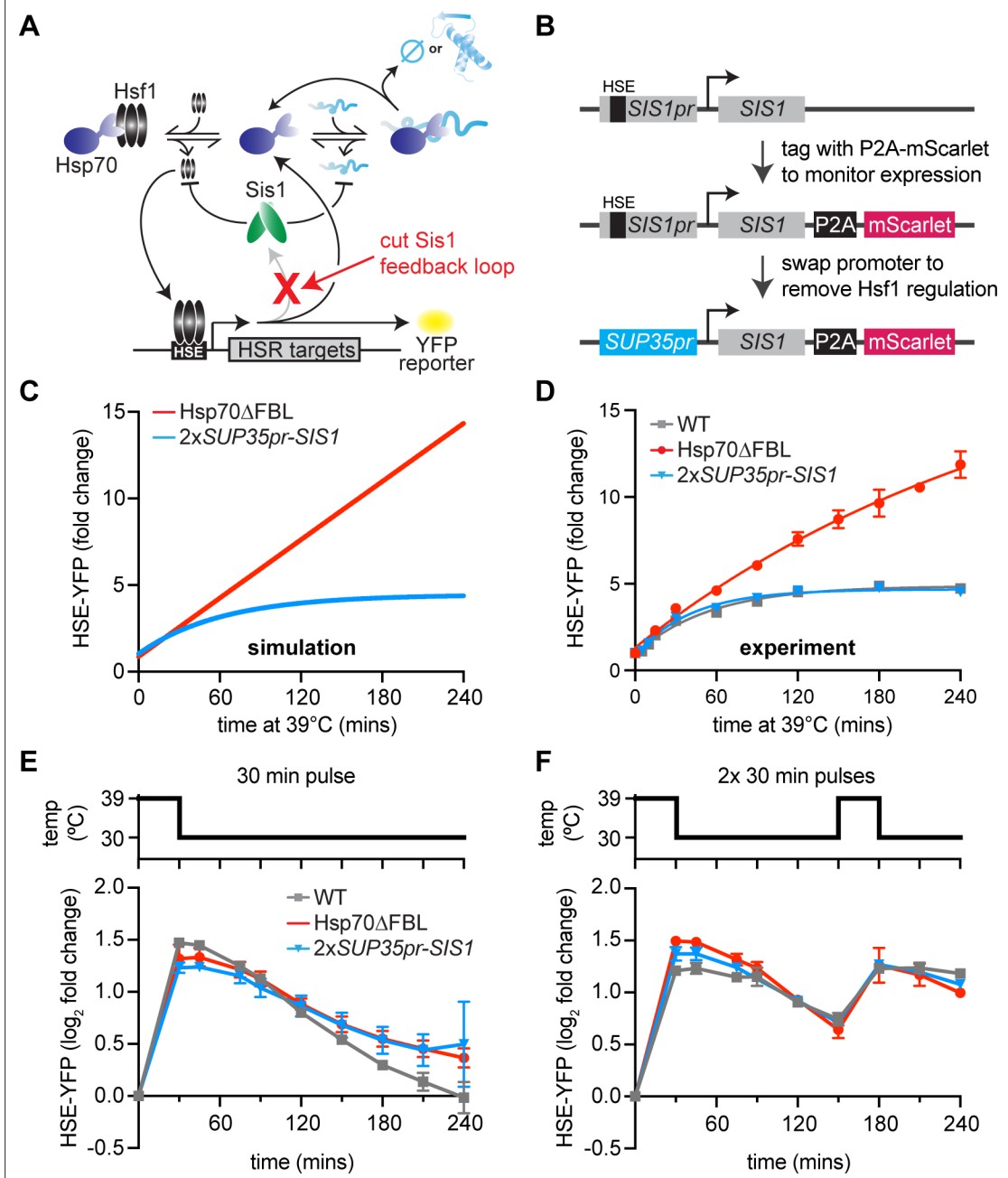

**Figure 3.** Transcriptional induction of Sis1 is dispensable for Hsf1 deactivation. (**A**) Network schematic. Sis1 induction by Hsf1 is cut to test whether Sis1 induction is necessary for normal Hsf1 regulation. (**B**) Generating the non-inducible Sis1 mutant. mScarlet fluorescent protein reports on Sis1 transcription and the Sis1 promoter is replaced with a non-inducible promoter. (**C**) HSE-YFP levels over a heat shock time course, measured by flow cytometry. Each data point represents the average and standard deviation of three biological replicates. (**D**) Simulation of HSE-YFP when Sis1 or Hsp70 is maintained at basal expression level during heat shock. (**E**) HSE-YFP levels during a 30-min heat shock followed by recovery at 30°C. (**F**) HSE-YFP levels during two 30-min heat shock intervals, separated by a 2-hr recovery at 30°C.

The online version of this article includes the following figure supplement(s) for figure 3:

**Figure supplement 1.** Decoupling Sis1 from Hsf1 regulation.

following heat shock. Sis1 is expressed at a moderate basal level under non-heat shock conditions, and its transcription is induced during heat shock. Hsf1 is responsible for both maintaining the basal expression level of Sis1 and inducing it during heat shock (*Pincus et al., 2018*; *Solís et al., 2016*).

As we had done for Hsp70, we tested the role of Sis1 induction during HS by removing its Hsf1-dependent transcriptional induction, first computationally and then experimentally (*Figure 3A, B*). Notably, while the mathematical model requires Hsp70-dependent transcriptional feedback to reca-pitulate the wild-type response dynamics, it does not include transcriptional induction of Sis1. Thus, the model accurately captures HSR response dynamics without a requirement for Sis1-mediated tran-scriptional feedback, that is, the model predicts that Sis1 induction is not required for negative feed-back (*Figure 3C*).

Experimentally, to maintain the basal expression level of Sis1, we could not simply remove the Hsf1-binding site from the *SIS1* promoter since Hsf1 drives its basal expression. Thus, we replaced the *SIS1* promoter sequence with the *SUP35* promoter (*Figure 3B*). According to our previously published RNA-seq data, *SUP35* mRNA is expressed at approximately the same level as endogenous *SIS1* mRNA under basal conditions but is not transcriptionally induced during heat shock (*Pincus et al., 2018*). To determine whether the *SUP35* promoter behaved as we expected, we measured expression of Sis1 by fusing it to P2A-mScarlet (a ribosomal skip sequence that enables fluorescent readout without tagging the protein of interest) (*Souza-Moreira et al., 2018*; *Figure 3B*). As expected, mScarlet was robustly induced during heat shock when expressed from the endogenous *SIS1* promoter, but it showed no induction by heat shock when expressed from the *SUP35* promoter (*Figure 3—figure supplement 1A*). However, we found that *SUP35*pr-*SIS1* cells displayed a moderate growth defect under non-stress conditions in a dilution series spot assay, and basal mScarlet expression from the *SUP35* promoter was reduced to half of the level expressed from the endogenous *SIS1* promoter (*Figure 3—figure supplement 1B, C*). To remedy this, we integrated a second copy of *SUP35*pr-*SIS1* into the *trp1* locus. In this 2x*SUP35*pr-*SIS1* strain, cell growth is comparable to wild type in a dilution series spot assay, and Hsf1 basal activity matches that of wild type (*Figure 3—figure supplement 1C, D*). Thus, the 2x*SUP35*pr-*SIS1* strain maintains the initial Sis1 expression level of wild-type cells while preventing induction of Sis1 during heat shock.

## Transcriptional induction of Sis1 is dispensable for Hsf1 deactivation

To determine whether Sis1 induction plays a role in feedback regulation of Hsf1, we monitored Hsf1 activity with the HSE-YFP fluorescent reporter. If Sis1 induction were necessary for Hsf1 deactivation, we would expect elevated Hsf1 activity during heat shock, similar to the strain we previously engi-neered to decouple Hsp70 expression from Hsf1 (Hsp70ΔFBL, for Hsp70 'deleted for the feedback loop') (*Figure 3D*). Briefly, in the Hsp70ΔFBL strain, all four paralogous genes encoding cytonuclear Hsp70 in yeast (*SSA1*, *SSA2*, *SSA3*, and *SSA4*) were deleted, and two copies of *SSA2* expressed from the constitutive and Hsf1-independent *TEF1* were inserted in the genome. The resulting strain expresses Hsp70 to wild-type levels under non-stress conditions but cannot induce more Hsp70 upon heat shock. In contrast to the prolonged induction of HSE-YFP displayed by Hsp70ΔFBL cells, 2x*SUP-35*pr-*SIS1* cells showed Hsf1 activity dynamics indistinguishable from wild type during heat shock (*Figure 3D*). These data suggest that, despite its role in repressing Hsf1 under basal conditions, Sis1 induction is dispensable for Hsf1 deactivation following heat shock, just as the model predicted.

Since Sis1 induction by Hsf1 is not necessary for Hsf1 deactivation following a single, sustained heat shock treatment, we wondered whether Sis1 induction might be necessary for deactivation of Hsf1 in more dynamic conditions. To test this, we subjected wild type, Hsp70ΔFBL and 2x*SUP35*-*SIS1* cells to various heat shock protocols and measured HSE-YFP dynamics. As we observed with the sustained heat shock, Hsf1 activity in non-inducible Sis1 cells was similar to wild type after a 30-min heat shock pulse followed by recovery at 30°C (*Figure 3E*). Moreover, HSE-YFP dynamics were also compa-rable to wild type when non-inducible Sis1 cells were exposed to a 2-hr heat shock pulse at 39°C followed by recovery at 30°C (*Figure 3—figure supplement 1E*). Interestingly, Hsp70ΔFBL showed Hsf1 activity comparable to wild type after the 30-min heat shock, suggesting that Hsp70 induction is less important for rapid Hsf1 deactivation than during sustained stress (*Figure 3E*). By contrast – and consistent with its hyperactive phenotype following sustained heat shock – Hsp70ΔFBL showed defec-tive Hsf1 deactivation after the 2-hr heat shock pulse (*Figure 3—figure supplement 1E*).

As a final test for defects in Hsf1 regulation in the non-inducible Sis1 lines, we exposed cells to multiple heat shock pulses with two different recovery periods: [30-min pulse–15-min recovery–30-min pulse] or [30-min pulse–2-hr recovery–30-min pulse]. With the longer 2-hr recovery period, both *2xSUP-35pr-SIS1* and Hsp70ΔFBL responded with similar induction dynamics to wild type (*Figure 3F*). This suggests that the 2-hr recovery time was sufficient for the cells to fully reset and respond equivalently to the second pulse. Following the shorter 15-min recovery period, Hsp70ΔFBL displayed increased Hsf1 activity relative to wild type, while *2xSUP35-SIS1* remained indistinguishable from wild type even in this challenging stress regime (*Figure 3—figure supplement 1F*). These data demonstrate that Sis1 induction does not serve to tune Hsf1 activity across any of the heat shock protocols we tested.

## Induced Sis1 is produced slowly and preferentially localizes to the nucleolar periphery

Given that Sis1 is not necessary for Hsf1 deactivation, we wanted to visualize the heat shock-induced protein to determine its localization. We've previously shown that Sis1 localized in the nucleolar periphery – and away from Hsf1 – upon activation of the HSR (*Feder et al., 2021*). If newly synthesized Sis1 were to likewise localize at the nucleolar periphery, it would not be in physical proximity to Hsf1, explaining its dispensability in the negative feedback loop.

To enable visualization of only newly induced Sis1, we tagged Sis1 with the HaloTag (*Los et al., 2008*) in a strain in which the nucleolar marker Nsr1 is tagged with mScarlet. The HaloTag irreversibly binds to synthetic ligands that contain a haloalkane moiety that can be attached to fluorescent dyes or remain non-fluorescent and block subsequent labeling. To image protein induced during heat shock only, we first incubated cells with the non-fluorescent ligand 7-bromoheptanol to block HaloTag conjugation to pre-existing Sis1 (*Figure 4A*; *Merrill et al., 2019*). Then we pulse labeled with the fluorescent ligand JF646 and started the heat shock simultaneously: JF646 thus exclusively marked heat shock-induced Sis1. We imaged newly synthesized Sis1-Halo over a heat shock time course and quantified both the induction dynamics and localization with respect to the Nsr1 nucleolar marker. Sis1 induction was undetectable until 10 min after heat shock and then increased linearly until 20 min of heat shock when it reached a plateau (*Figure 4B and C*). Moreover, once it became visible, newly synthesized Sis1 preferentially localized adjacent to the nucleolus as determined by a sustained increase in the proportion of Sis1 surrounding the Nsr1 signal (*Figure 4B, D*). Next, we imaged newly synthesized Sis1-Halo in cells expressing Hsf1-mVenus. As recently reported, Hsf1 forms subnuclear condensates during heat shock, and we observe little colocalization between these Hsf1 foci and newly synthesized Sis1 (*Figure 4E, F*; *Chowdhary et al., 2022*). These data suggest that newly synthesized Sis1 is not spatially positioned to repress Hsf1 and inactivate the HSR immediately following heat shock.

We repeated the pulse-labeling analysis for Ssa1-Halo – the *SSA1* gene encodes an inducible Hsp70 – as we previously showed that Hsp70 induction is required for HSR deactivation (*Krakowiak et al., 2018*). In contrast to Sis1, we were able to detect newly synthesized Ssa1 after only 6 min of heat shock, and Ssa1 was induced substantially more than Sis1 by 15 min (*Figure 4B, C*). Moreover, induced Ssa1 localized to the nucleolar periphery as well as to puncta and more diffusely throughout the cell (*Figure 4B*). Newly synthesized Ssa1 was less concentrated at the nucleolar periphery over the heat shock time course than newly synthesized Sis1 (*Figure 4D*). Moreover, in contrast to Sis1, newly synthesized Ssa1-Halo partially colocalized with Hsf1 (*Figure 4E, F*). The combination of the delayed Sis1 induction relative to Ssa1 and the localization of Sis1 primarily to the nucleolar periphery – and away from Hsf1 – may explain why Sis1 is not a feedback regulator of the HSR.

## Sis1 transcriptional regulation confers fitness in nonfermentable carbon sources

Without an obvious role in negative feedback regulation, we investigated whether Sis1 transcriptional regulation by Hsf1 promotes fitness. We quantitatively monitored fitness by comparing optical density of the *2xSUP35pr-SIS1* strain to the Hsp70ΔFBL strain and wild type in culture using an automated plate reader. Cells were grown for 4 hr at 30°C, then were either kept at 30°C or shifted to 37°C. All strains showed comparable growth at 30°C (*Figure 5—figure supplement 1A*). However, while Hsp70ΔFBL was indistinguishable from wild type at 37°C, the *2xSUP35pr-SIS1* strain showed reduced fitness compared to the other strains after the first 4 hr at 37°C and failed to undergo a diauxic shift

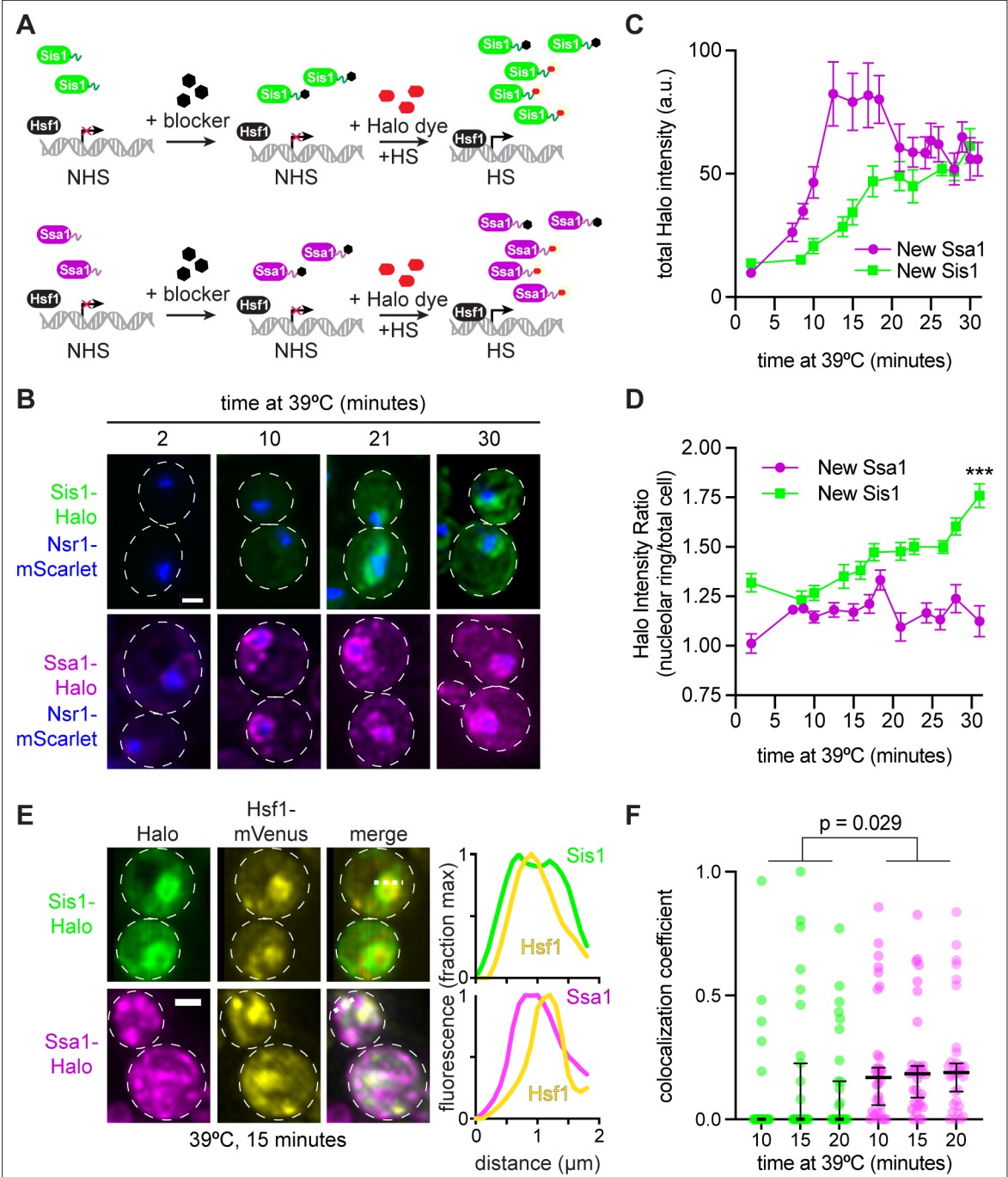

**Figure 4.** Sis1 is induced slowly and localizes away from Hsf1 following heat shock. (**A**) Experiment workflow. Cells were treated with 20 μM 7-bromoheptanol for 5 min, then incubated in 20 μM of JF646 Halo dye at 39°C. Halo fluorescence thus marks the population of Sis1 or Hsp70 translated during heat shock. (**B**) Representative deconvolved live cell images captured by lattice light sheet microscopy. Cells imaged after 2-,10-, 21-, and 30-min heat shock. Cells express Nsr1-mscarlet as well as Sis1-Halo or Ssa1-Halo dye. Scale bar represents 2 μm. (**C**) Quantification of total cell Halo intensity during heat shock time course. Each point represents the mean of the average Halo intensity across 5–20 cells. Error bars represent the standard error. (**D**) Quantification of mean Halo intensity at the nucleolar periphery divided by mean Halo intensity in the total cell. Each data point represents the average ratio across 5–20 cells. Error bars indicate the standard error. ***, p<0.001 by two tailed t-test. (**E**) (Left) Representative deconvolved live cell images captured by lattice light sheet microscopy of cells expressing Hsf1-mVenus and newly synthesized Sis1 or Ssa1. (Right) Normalized fluorescence intensity along the dotted line shown in the images. (**F**) Single-cell analysis of the fraction of total newly synthesized Sis1 (green) or Ssa1 (magenta) that colocalizes with Hsf1. Lines are at the median, and error bars depict 95% confidence intervals.

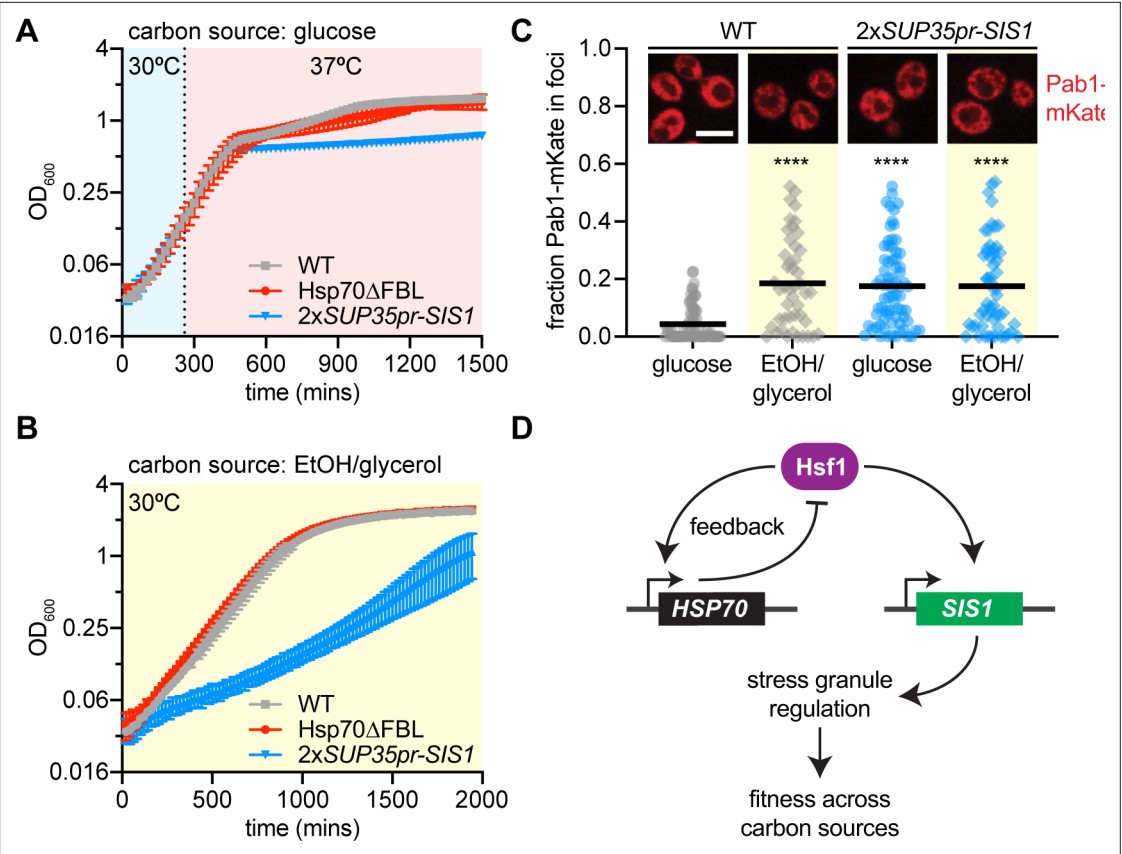

**Figure 5.** Sis1 transcriptional regulation confers fitness under stress. (**A**) Quantitative growth curve of non-inducible strains during heat shock (37°C). (**B**) Quantitative growth curve of non-inducible strains during growth in limited carbon media (2% glycerol, 2% EtOH) in non-heat shock (30°C). (**C**) Single-cell quantification of the fraction of Pab1-mCherry localized in foci as determined by the FindFoci plugin in Fiji. Inset: Representative deconvolved live cell images captured by lattice light sheet microscopy of cells expressing Pab1-mKate. (**D**) Model of the differential roles of transcriptional regulation of Hsp70 and Sis1 in feedback and fitness, respectively.

The online version of this article includes the following figure supplement(s) for figure 5:

**Figure supplement 1.** Additional growth curves.

once glucose was depleted from the media (*Figure 5A*). This suggests that induction of Sis1 may play a role in promoting growth in the absence of a fermentable carbon source.

To directly test whether Hsf1-dependent regulation of Sis1 promotes growth in nonfermentable media, we grew cells with EtOH/glycerol as the carbon source at 30°C. Indeed, the *2xSUP35pr-SIS1* strain grew substantially slower than wild type or Hsp70ΔFBL in EtOH/glycerol media (*Figure 5B*). Remarkably, the 1x*SUP35pr-SIS1* strain – which has a slow growth phenotype in glucose (*Figure 3— figure supplement 1C*) – completely rescues growth in EtOH/glycerol (*Figure 5—figure supplement 1B*), suggesting that the 2x*SUP35pr-SIS1* strain grows slowly on nonfermentable carbon due to the inability to downregulate Sis1 rather than induce it. Consistent with this interpretation, when we increased Sis1 expression even more by adding a second copy of Sis1 under its stronger endogenous promoter, the growth phenotype in EtOH/glycerol was exacerbated to the point that cells failed to grow altogether (*Figure 5—figure supplement 1B*). Thus, it is not induction per se but the ability to modulate Sis1 expression level according to the metabolic state of the cell that appears to promote fitness.

## Transcriptional regulation of Sis1 coordinates stress granules with carbon metabolism

In addition to regulating Hsf1 activity under basal conditions in the nucleus, Sis1 is known to regulate cytosolic stress granules. Stress granules are biomolecular condensates composed predominately of

mRNA and translation factors that form in yeast response to a variety of stressors including both heat shock and glucose depletion. To determine whether stress granules could provide a link between Sis1 regulation and fitness in media lacking glucose, we imaged the stress granule marker Pab1 fused to the mKate fluorescent protein in wild-type cells and 2x*SUP35pr-SIS1* cells. Pab1-mKate was diffuse throughout the cytosol in wild-type cells in glucose media but a fraction formed small granules in EtOH/glycerol media; by contrast, Pab1-mKate was granular in both glucose and EtOH/glycerol in 2x*SUP35pr-SIS1* cells (*Figure 5C*). These results suggest there are constitutive stress granules in 2x*SUP35pr-SIS1* cells, connecting a known activity of Sis1 in regulating stress granules with its apparent role in carbon metabolic homeostasis. This result suggests that proper regulation of Sis1 expression is required to dynamically regulate stress granules, conferring fitness as carbon source fluctuates in the environment.

## Discussion

In this study, we formally incorporated two recent findings into a mathematical model of the HSR: (1) the role of NSPs as a major class of molecular species that drive HSR activation (*Masser et al., 2019*; *Tye and Churchman, 2021*), and (2) the role of the JDP Sis1 in repressing HSR activity under non-stress conditions by promoting the interaction of Hsp70 with Hsf1 (*Feder et al., 2021*). Importantly, while the addition of these new features makes the model more realistic and enabled accurate recapitulation of experimental perturbations to protein synthesis and Sis1, our results indicate that the core circuitry of the HSR that we previously proposed need not be embellished: the heart of the HSR remains a two-component feedback loop between Hsp70 and Hsf1.

Our previous model, as well as those of other groups, assumed that the HSR is triggered by the denaturation of mature proteins (*Feder et al., 2021*; *Krakowiak et al., 2018*; *Petre et al., 2011*; *Scheff et al., 2015*; *Zheng et al., 2016*). These models simulated the initiation of stress as the instantaneous accumulation of a single bolus of heat-induced unfolded or misfolded proteins that required chaperones for triage, refolding, and degradation. However, these models are unable to account for the reduction in HSR output observed in heat-shocked cells that had been treated with protein synthesis inhibitors or otherwise cultured in growth conditions with reduced protein synthesis (*Masser et al., 2019*; *Triandafillou et al., 2020*; *Tye and Churchman, 2021*; *Tye et al., 2019*). To reflect the current understanding that the HSR is triggered, at least in part, by NSPs, we explicitly model translation and subsequent protein maturation/folding. Upon heat shock, the elevated temperature decreases the protein folding rate, thereby resulting in the accumulation of immature NSPs. These NSPs in turn titrate Hsp70 and Sis1, thereby activating the HSR. To test this mechanism, we experimentally decreased the pool of NSPs by pretreating cells with rapamycin prior to heat shock. Indeed, compared to untreated cells, we observed decreased HSR output in the rapamycin-treated cells following heat shock in the model and in cells, validating the NSP-based activation mechanism. In addition to NSPs, it has been proposed that heat-induced biomolecular condensates of mature proteins and mRNAs may activate the HSR (*Triandafillou et al., 2020*). If/when these putative condensate agonists are identified, they will be incorporated into future iterations of the model.

Regarding the HSR feedback loop, we previously found that in addition to repressing the HSR under basal conditions, induction of Hsp70 is required to deactivate the HSR following a sustained heat shock (*Krakowiak et al., 2018*). Thus, Hsp70 is a negative feedback regulator of the HSR. Given the role of Sis1 in repressing the HSR in the absence of stress and the fact that it is also a transcriptional target of the HSR, we expected that Sis1 too would provide negative feedback. However, after decoupling Sis1 expression from the HSR, we observed no difference in HSR activation dynamics compared to wild-type cells, demonstrating that Sis1 induction is dispensable for deactivation of the HSR. Since induction of Hsp70 is required for deactivation of the HSR, it is possible that the increased levels of Hsp70 obviate the requirement for Sis1. Alternatively, the basal concentration of Sis1 may be sufficient to promote Hsf1 inactivation, or induction of another JDP may be required.

Though dispensable for Hsf1 deactivation, we found that Sis1 induction was necessary for long-term fitness during heat shock and in nonfermentable carbon sources, and we linked this to a disruption in the regulation of stress granules (*Figure 5*). Sis1 plays a critical role in disaggregation of stress-induced biomolecular condensates in cooperation with Hsp70 and Hsp104 (*Yoo et al., 2022*), and our results here imply that regulation of Sis1 levels may in turn regulate the condensates that form during heat shock or glucose depletion. In addition to cytosolic stress granules, Sis1 is also targeted to the

periphery of the nucleolus during heat shock, where it interacts with accumulated ribosomal proteins that are known to activate the HSR when produced in stoichiometric excess to ribosomal RNA (*Albert et al., 2019*; *Feder et al., 2021*; *Tye et al., 2019*). The peri-nucleolar localization of induced Sis1 we observed here in the pulse-labeling experiments (*Figure 4B, D*) is consistent with a role for Sis1 as an orphan ribosomal protein chaperone (*Ali et al., 2022*). The emerging role of Sis1 in regulating adaptive stress-induced condensates may be among its essential functions.

Our experimental findings imply that Hsp70 induction is still the principal source of negative feedback for the HSR, and that the mechanism of HSR repression during basal conditions may be distinct from the deactivation mechanism following stress. Further investigations should systematically explore whether Hsp70 is unique in its role as a negative feedback regulator or if other Hsf1 targets confer feedback to the HSR either independent of or via Hsp70. With an input function based on the accumulation of NSPs and the inclusion of Sis1 in the regulatory circuitry, the new iteration of the model presented here brings us closer to a mechanistically accurate and predictive understanding of the HSR.

## Methods
### Mathematical modeling
To model the Hsp70–Hsf1 feedback circuit we have expanded the previous model (*Feder et al., 2021*) to explicitly consider the production of unfolded nascent proteins that are folded into their native state with some rate. More specifically, the model consists of four different protein species (Hsf, Hsp, UP, reporter YFP) and two complexes (one between Hsp and Hsf, and the other between Hsp and UP). The differential equations describing the cellular concentration of these species are as follows:

$$\frac{d\left[HSP\right]}{dt} = k_2\left[HSP\cdot Hsf\right] - k_1\left[HSP\right]\left[Hsf\right] + \left(k_4+k_5\right)\left[HSP\cdot UP\right] - k_3\left[HSP\right]\left[UP\right] + \beta\frac{\left[Hsf\right]^n}{K_d^n + \left[Hsf\right]^n}$$

$$\frac{d\left[Hsf\right]}{dt} = k_2\left[HSP\cdot Hsf\right] - k_1\left[HSP\right]\left[Hsf\right]$$

$$\frac{d\left[UP\right]}{dt} = k_{up} - k_{dup}\left[UP\right] + k_4\left[HSP\cdot UP\right] - k_3\left[HSP\right]\left[UP\right]$$

$$\frac{d\left[HSP\cdot HSF\right]}{dt} = k_1\left[HSP\right]\left[Hsf\right] - k_2\left[HSP\cdot Hsf\right]$$

$$\frac{d\left[HSP\cdot UP\right]}{dt} = k_3\left[HSP\right]\left[UP\right] - \left(k_4+k_5\right)\left[HSP\cdot UP\right]$$

$$\frac{d\left[YFP\right]}{dt} = \beta\frac{\left[Hsf\right]^n}{K_d^n + \left[Hsf\right]^n},$$

where [] denotes the cellular concentration of respective species. The interpretation of all models here is identical to the original model (*Zheng et al., 2016*), except for two new parameters – the de novo synthesis of unfolded proteins $k_{up} = 1\ min^{-1}a.u.^{-1}$ and their folding with rate $k_{dup} = 0.35\ min^{-1}$. The rate $k_1 = 320\ min^{-1}a.u.^{-1}$ denotes the binding of Hsp to Hsf to create an inactive complex $HSP\cdot Hsf1$, and the complex dissociates with rate $k_2 = 0.7\ min^{-1}$. The rate $k_3 = 112\ min^{-1}a.u.^{-1}$ is the binding of Hsp to unfolded proteins (denoted as $UP$) to create the complex $HSP\cdot UP$ that dissociates with rate $k_4 = 0.1\ min^{-1}$. The degradation of UP by Hsp is captured via the rate $k_5 = 0.15\ min^{-1}$. The activation of both YFP and Hsp by Hsf is modeled by a Hill equation with $n = 3$, $\beta = 0.1\ min^{-1}$ and $k_d = 0.0025\ a.u.$
The above differential equation model was run with the following initial values (in a.u.) at time $t = 0$

$$\left[HSP\right] = \frac{1}{2},\ \left[Hsf1\right] = 0,\ \left[HSP\cdot HSF1\right] = \frac{1}{250},\ \left[UP\right] = 0,\ \left[HSP\cdot UP\right] = 0,\ \left[YFP\right] = 1.$$

The effect of rapamycin is captured by decreasing the rate $k_{up}$ and trajectories corresponding to different heat shock temperatures are obtained by perturbing the folding rate $k_{dup}$. To investigate the experiment-to-experiment fluctuations in YFP trajectories we solved the above system by varying the parameter $k_{up}$ around its nominal value by 20% at the start of the simulation. The impact of Sis1 depletion is captured by destabilizing the Hsp–Hsf complex via an increase in the dissociation rate $k_2$.

Finally, the removal of Hsp feedback is realized by removing the Hsf-induced activation term in the equation for $\frac{d[HSP]}{dt}$.

## Strain construction

Yeast strains and plasmids used in this study are listed in *Supplementary file 1*. All strains are derived from the W303 parent strain. CRISPR-mediated promoter swapping was performed to create the 1xSup35pr-Sis1 (1701) and 1xSup35pr-Sis1-P2A-mscarlet (1661) strains. CRISPR-Cas9 mediated precise, scarless replacement of the native Sis1 promoter with the 600 bp Sup35 promoter. To construct the final non-inducible Sis1 line (2xSup35pr-Sis1, 1761), we incorporated a second copy of Sup35pr-Sis1 at the tryp locus.

In all other cloned lines, genes were tagged at the endogenous locus. Cells were transformed with double-stranded DNA fragments containing ~20 bp homologous flanking regions. This method takes advantage of homology-directed repair mechanisms in *S. cerevisiae*, as described previously (*Longtine et al., 1998*).

## Cell growth

For heat shock time courses followed by flow cytometry, cells were cultured in 1xSDC media overnight at room temperature (synthetic media with dextrose and complete amino acids). Before RT-PCR or cell growth assays, cells were cultured in yeast extract peptone dextrose (YPD) media shaking at 30°C overnight. Cells were subjected to heat shock at 39°C unless otherwise specified.

## HSE-YFP reporter heat shock assays

Three biological replicates of each strain were serially diluted five times (1:5) in 1xSDC and grown overnight at room temperature. In the morning, cells had reached logarithmic phase, and 750 µl of each replicate was transferred to a PCR tube and shaken for 1 hr at 30°C to aerate. Then, cells were exposed to heat shock at 39°C. At the pre-determined time points (0, 5, 10, 15, 30, 60, 90, 120, 180, and 240 min), 50 µl of cell culture was transferred to a well of a 96-well plate, containing containing 1xSDC and a final concentration of 50 mg/ml CHX to stop translation. After the time course, cells were incubated at 30°C for 1 hr to allow fluorescent reporter maturation before measurement. All experiments were performed using C1000 Touch Thermal Cycler (Bio-Rad). Cell fluorescence was measured by flow cytometry and results were analyzed as described below.

## Anchor-away assay

Cells were grown overnight as described above. Rapamycin was added to a final concentration of 10 µM, and the time course was started immediately. During the time course, cells were maintained at 30°C shaking. At the predetermined time points (0, 15, 30, 45, 60, 75, and 90 min), 50 µl of cells were transferred to a 96-well plate identically to the heat shock assay described above.

## Translation inhibition assay (rapamycin pretreatment)

Cells were grown overnight, and in the morning, shaken at 30°C for 1 hr. Rapamycin was added to a final concentration of 10 µM, and cells were left shaking at 30°C for 5 min. Then, the heat shock time course was performed at 39°C as described above.

## Flow cytometry

HSE-YFP and mscarlet reporter levels in heat shock time course and rapamycin treatment assays were measured at the University of Chicago Cytometry and Antibody Technology Facility. These measurements were performed using the 488-525 FITC fluorescence filter (HSE-YFP) and 561-PE Dazzle (mscarlet) on the BD Fortessa High Throughput Flow Cytometer. The raw fluorescence values were normalized by side scatter in FlowJo. Then, the median fluorescence value was calculated. Each data point represents the average of three biological replicates.

## Dilution series spot growth assays

Yeast strains were grown overnight shaking at 30°C in YPD. In the morning, they were diluted to and final optical density (OD) of 1 and serially diluted 1:10 in water. Each diluted yeast culture was spotted onto 1xYPD plates. Images were taken after 2 days of growth at 30 or 37°C.

## Heat shock quantitative growth assay

Yeast strains were grown overnight at 30°C in 1xYPD. In the morning, they were diluted to OD = 0.1 in 1xYPD. SpectroStar Nano microplate reader was used to measure cell density every 20 min. Each data point represents the mean of three biological replicates.

## Glucose starvation quantitative growth assay

Yeast strains were grown for 24 hr in 2% glucose YEP media then diluted to OD = 0.1 and grown overnight again. In the morning, cells were diluted to OD = 0.1 in 1xYEP containing 2% glycerol, 2% EtOH and growth was monitored every 20 min at 30°C.

## RNA quantification and analysis

RT-qPCR was performed as previously described in *Chowdhary et al., 2022*, except for the following: cells were grown at 30°C in YPD to a mid-log density ($OD_{600}$ = 0.8). Rapamycin was added to a final concentration of 10 µg/ml for 5 min. At this point, a portion of the culture was maintained at 30°C (0 min HS) and the remainder was subjected to heat shock at 39°C for 5, 15, 60, and 120 min. Cells were flash frozen immediately after heat shock, and then harvested. Total RNA was extracted using hot acid phenol extraction method, followed by ethanol precipitation (*Solís et al., 2016*). To determine fold change mRNA levels, mean mRNA levels of heat-shocked samples were normalized to the NHS sample (*Pfaffl, 2001*).

> mVenus forward primer: caacattgaagatggtggtgttc
> mVenus reverse primer: ctttggataaggcagattgatagg

## RNA-seq analysis

RNA read counts for 39 Hsf1 target genes were collected in a recent paper and reanalyzed here (*Triandafillou et al., 2020*; GEO accession number GSE152916).

Experimental methods are described in detail in the original paper. In brief, cells were exposed to heat shock at 42°C for 20 min. Cells exposed to heat shock and CHX were treated with 200 µg/ml CHX simultaneously upon HS. Each data point represents average read count for a single Hsf1 target gene (two biological replicates). Translation dependence was calculated as 1 − (CHX + HS/HS).

## Halo-Tagging

To image NSPs, we incubated cells with blocker (20 µM, 7-bromoheptanol) for 5 min, washed two times with 2xSDC media and then incubated with 0.4 µM of Halo JF646dye at 39°C, to start HS treatment simultaneously. Under this treatment, only protein induced during heat shock was visualizable. Lattice light sheet microscopy was used to visualize 5–20 cells every 2.5 min throughout the 30-min heat shock.

## Lattice light sheet microscopy and quantification

Lattice light sheet imaging was performed at the University of Chicago Integrated Light Microscopy Core (Intelligent Imaging Innovations) and run in SlideBook 6.0 software. Captured images were deconvoluted using Graphics processing unit-based Richardson–Lucy deconvolution with measured PSFs via Brian Northan's 'Ops' implementation (https://github.com/imagej/ops-experiments; *Northan, 2022*). 3D reconstructions and videos were assembled using ClearVolume (*Royer et al., 2015*). The mean Halo intensity at the ring around the Nsr1-marked nucleolus was divided by the mean intensity over the total cell.

## Acknowledgements

We thank Arvind Murugan and Kabir Husain for use of the cell culture plate reader and the University of Chicago Integrated Light Microscopy facility for use of the lattice light sheet microscope. We also thank Alex Ruthenburg for helpful suggestions and members of the Pincus lab for insightful discussions and critical reading of the manuscript.

## Additional information

### Funding

| Funder | Grant reference number | Author |
|---|---|---|
| National Institutes of Health | GM124446 | Abhyudai Singh |
| National Science Foundation | OMA-2121044 | David Pincus |
| National Institutes of Health | GM138689 | David Pincus |

The funders had no role in study design, data collection, and interpretation, or the decision to submit the work for publication.

### Author contributions

Rania Garde, Conceptualization, Resources, Formal analysis, Investigation, Visualization, Methodology, Writing - original draft, Writing - review and editing; Abhyudai Singh, Conceptualization, Resources, Formal analysis, Funding acquisition, Investigation, Methodology, Writing - review and editing; Asif Ali, Resources, Formal analysis, Investigation, Methodology, Writing - review and editing; David Pincus, Conceptualization, Supervision, Funding acquisition, Visualization, Methodology, Writing - original draft, Writing - review and editing

### Author ORCIDs

Abhyudai Singh ⓘ http://orcid.org/0000-0002-1451-2838
David Pincus ⓘ http://orcid.org/0000-0002-9651-6858

### Decision letter and Author response

Decision letter https://doi.org/10.7554/eLife.79444.sa1
Author response https://doi.org/10.7554/eLife.79444.sa2

## Additional files

### Supplementary files

• Supplementary file 1. Table of yeast strains used in this study. Associated with all figures.

• MDAR checklist

### Data availability

All data presented in the paper and custom analysis software are deposited in Dryad and Zenodo, respectively, with the following DOIs: https://doi.org/10.5061/dryad.b2rbnzsm6, https://doi.org/10.5281/zenodo.7860686.

The following datasets were generated:

| Author(s) | Year | Dataset title | Dataset URL | Database and Identifier |
|---|---|---|---|---|
| Garde R, Singh A, Ali A, Pincus D | 2023 | Induction of Sis1 promotes fitness but not feedback in the heat shock response | https://doi.org/10.5061/dryad.b2rbnzsm6 | Dryad Digital Repository, 10.5061/dryad.b2rbnzsm6 |
| Garde R, Singh A, Ali A, Pincus D | 2023 | Induction of Sis1 promotes fitness but not feedback in the heat shock response | https://doi.org/10.5281/zenodo.7860686 | Zenodo, 10.5281/zenodo.7860686 |

The following previously published dataset was used:

| Author(s) | Year | Dataset title | Dataset URL | Database and Identifier |
|---|---|---|---|---|
| Triandafillou CG | 2020 | Measuring the dependence of the yeast heat shock response on intracellular pH and translation during stress | https://www.ncbi.nlm.nih.gov/geo/query/acc.cgi?acc=GSE152916 | NCBI Gene Expression Omnibus, GSE152916 |

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
