## [Editor Report]

Here the authors describe an updated theoretical model describing how the Hsf1 transcription factor is activated in yeast in response to heat shock, and then test the new model experimentally, providing solid evidence that heat shock results in delayed folding of newly synthesized proteins (NSPs), which sequester the Hsp70 chaperone away from the inactive Hsp70/Hsf1 complex, releasing active Hsf1. They also demonstrate convincingly that the Hsp70 co-chaperone Sis1 is not required directly for the heat shock response (HSR), but under basal conditions targets Hsf1 to Hsp70 for repression and then upon upregulation by Hsf1 serves to promote fitness by coordinating stress granule and carbon metabolism to improve fitness in the presence of nonfermentable carbon sources. By taking into account NSPs and incorporating new roles for the Sis1 co-chaperone in their updated model, these studies represent a significant advance for the heat shock response field.

---

## [Decision Letter]

**Decision letter after peer review:**

Thank you for submitting your article "Induction of Sis1 promotes fitness but not feedback in the heat shock response" for consideration by *eLife*. Your article has been reviewed by 3 peer reviewers, including Tony Hunter as the Reviewing Editor and Reviewer #1, and the evaluation has been overseen by Jonathan Cooper as the Senior Editor. The following individuals involved in review of your submission have agreed to reveal their identity: Claes Andréasson (Reviewer #2); Kevin A Morano (Reviewer #3).

The three reviewers agree that your studies provide a needed update and validation of your earlier mathematical model that describes HSR/Hsf1 regulation in response to heat shock in yeast, and provide new insights into how Hsf1 transcriptional activity is regulated in response to heat shock. First, your new findings show that upon heat shock it is newly synthesized proteins rather than denatured mature proteins that sequester the Hsp70 chaperone away from Hsf1, thus permitting Hsf1 to bind to target genes and drive the heat shock response (HSR) gene transcription program. Second, evidence is provided that the Sis1 Hsp70 co-chaperone does not have a direct negative role in the HSR, but rather is needed for fitness during prolonged stress. Both the impact of newly translated proteins as negative regulators of Hsp70 Hsf1 chaperone activity during heat shock and an indirect role for Sis1 in the HSR have been incorporated as new nodes in the updated model, which is shown experimentally to be able to recapitulate kinetically the observed cellular responses to heat shock at different temperatures. However, the reviewers have raised some issues that need to be addressed in a revised version.

Essential revisions:

Three essential issues need to be addressed with new experiments before submission of a revised version.

1. If possible, more needs to be done to better define the subnuclear localization of Sis1 with respect to both Hsf1 and Ssa1 to establish that the location of Sis1 is physically distinct from that of Ssa1, and that in consequence Sis1 is not in a position to modulate Ssa1 chaperone activity directly.

2. Stronger evidence is needed to establish whether Sis1 has the proposed general role in maintaining cellular fitness under stress conditions, and ideally some mechanistic insights provided into this role.

3. Additional controls are needed to establish that the 2XSUP35prom-SIS1 strain does not have unlinked respiratory defects that might account for some of the observed phenotypes.

Some additional specific suggestions for improvement can be found in the individual reviews. We look forward to receiving a revised version.

*Reviewer #1 (Recommendations for the authors):*

The two main weaknesses are the lack of insight into how Sis1 is acting to promote fitness under heat shock conditions, and whether sequestration of Sis1at the nucleolar periphery is really a viable mechanism for preventing its actin in the nucleus, or whether some other event such as an inactivating post-translational modification of Sis1 under heat shock conditions might be involved. Further experiments to address these issues are recommended.

*Reviewer #2 (Recommendations for the authors):*

1. The experimental result in Figure 2B shows that Hsf1 activity increases linearly during the entire time course (90 min) following Sis1 depletion. No plateau is reached within this time-frame. Yet, the model in Figure 2C predicts that a plateau should be reached. This impacts on the usefulness and how faithfully the model recapitulates Hsf1 regulation. Explaining the discrepancy between experiment and model would strengthen the paper. (Perhaps Sis1 depletion results in collapse of the proteostasis network and indirect effects due to massive protein misfolding takes over, which makes interpretation of later time points not very meaningful.)

2. In Figure 2D experimental variation in Hsf1 activity after heat-shock is documented and is the simulated in Figure 2E by altered rates of protein synthesis in the model. The authors claim that initial metabolic states of the cells are causing the experimental variation by impacting on translation (lines 192-204). I suggest to directly test this hypothesis by simply applying stricter conditions during microbiological culturing. In the Methods section (Lines 600-605) it is stated that cells were cultured at room temperature and that over-night cultures were used. Cells from over-night cultures may have variable metabolic states due to variable nutrient (glucose) consumption and room temperature may or may not be stable adding to the variability. Lines 608-611 describe the use of a mild 1 h pre-heat-shock (30 degree C) in a PCR tube with a relatively large volume of culture (750 microL). This raises the question if cells even get enough oxygen under these conditions? Repeating the experiment with cultures grown to log phase from an overnight culture for at least three generations (OD600=0.1 to = 0.8) at a stable temperature provides a test to the claim that the variability comes from the metabolic state of the cultures, i.e. experimental variation should decrease if the hypothesis is true.

3. The authors show that the 2XSUP35pr-SIS1 strain does not undergo diauxic shift and grows poorly on non-fermentable carbon sources (Figure 5A-B, lines 327-346). This finding is consistent with the claim that Sis1 induction by Hsf1 promotes fitness in the presence of non-preferred carbon sources. Yet the SIS1 promoter is also regulated by Msn2 and Msn4 that are activated when cells grow on non-preferred carbon sources. It may be Msn2 and Msn4 and their binding sites that explain the phenotype rather than decoupling from Hsf1 regulation. It would be valuable to distinguish between these possibilities so the study accurately reflects the involvement of Hsf1/HSE in the phenotype.

4. A concern I have is that the 2XSUP35pr-SIS1 strain has lost respiratory capacity during strain construction and that this phenotype is not genetically linked to the SIS1 promoter replacement. The appearance of respiratory deficient phenotypes is quite common when transforming yeast cells and the strain has undergone several sequential steps of genetic modification, some of which include slow growth. In light of the central part this strain plays in the study I strongly suggest ruling out this scenario by performing additional controls. For example, complementation of the respiratory phenotype with WT SIS1 or alternatively repair of the promoter could be potential routes. Strain crosses and analysis of the meiotic offspring is another way.

Recommended text changes

a) Lines 26-28. Rephrase "Unexpectedly". I got stuck on here when reading it for the first time. It is probably unclear for most readers why this is unexpected and I actually do not find it to be unexpected.

b) Lines 103. Rephrase "However" since the previous sentence is not contrasted. I got stuck on here when reading it for the first time.

c) Lines 227-253 contains a lengthy description of the construction and rationale behind the strain referred to as 2xSUP35pr-SIS1. I appreciate all the work behind the strain construction but the description adds little to the Results. I suggest editing out this part and moving any needed parts to the Methods section so that the paragraphs starting at line 216 and 255, respectively, become merged to one single paragraph.

*Reviewer #3 (Recommendations for the authors):*

1. It is unclear how the misfolding rates (Kdup) at different temperatures are calculated and modeled. Clearly, the experimental results appear to mirror the modeling output but how the authors got there is not apparent.

2. The Hsp70∆FBL strain is not explained in the text and will be confusing to readers unfamiliar with previous work from this group.

3. The Sis1 localization experiments in Figure 4B appear to show significant nuclear localization rather than exclusively nucleolar peripheral signal. It might be useful to include a DAPI stain or nucleoplasmic tagged protein to conclusively state that Sis1 is nucleolar.

4. Also, in Figure 4B, I expected to see Ssa1 exhibit nucleoplasmic localization in addition to cytoplasmic puncta. At the resolution provided, it's hard to discern any significant difference between Sis1 at the nuclear/nucleolar level, yet the results are interpreted in opposite directions – Sis1 is not where it needs to be to regulate Hsf1, but Ssa1 is. Is it possible to perform ChIP experiments to discern whether Sis1 is present at HSEs with Hsp70 and Hsf1 during the attenuation phase?

5. As stated above, it is not clear what the reason is for the post-diauxic shift and ethanol/glycerol growth phenotypes. Sis1 is an essential gene, so it is not surprising that perturbations in expression might lead to growth defects under non-ideal conditions. To claim this is an important "fitness" outcome related to Hsf1-mediated control of SIS1 gene expression is an overinterpretation of the data. The fact that the Hsp70∆FBL strain exhibits no such problems argues against a need for precise Hsf1 regulation under these growth conditions.

---

## [Author Response]

Essential revisions:Three essential issues need to be addressed with new experiments before submission of a revised version.1. If possible, more needs to be done to better define the subnuclear localization of Sis1 with respect to both Hsf1 and Ssa1 to establish that the location of Sis1 is physically distinct from that of Ssa1, and that in consequence Sis1 is not in a position to modulate Ssa1 chaperone activity directly.

To address this concern, we constructed two new imaging strains expressing Hsf1-mVenus/Halo-Sis1 and Hsf1-mVenus/Halo-Ssa1 (Hsp70) and used pulse-labeling followed by live lattice light sheet 3D imaging to resolve the subcellar localization of newly synthesized Sis1 and Hsp70 with respect to Hsf1 over a heat shock time course. Unfortunately, we cannot monitor newly induced Sis1 and newly induced Hsp70 simultaneously in the same cells with the HaloTag pulse labeling system. We found that a significantly greater fraction of newly synthesized Hsp70 colocalizes with Hsf1 than new Sis1. Thus, while we cannot directly image new Sis1 and Hsp70 in the same cell, we clearly observe a differential localization pattern with respect to Hsf1. These data are included in the revised Figure 4.

2. Stronger evidence is needed to establish whether Sis1 has the proposed general role in maintaining cellular fitness under stress conditions, and ideally some mechanistic insights provided into this role.

We have now connected the growth phenotypes of non-inducible Sis1 in the diauxic shift and in EtOH/glycerol media to mis-regulation of stress granules as marked by Pab1-mScarlet. Under nonstress conditions in glucose-containing media in wild type cells, Pab1-mScarlet is mostly diffuse in the cytosol. However, glucose depletion is known to drive Pab1 into cytosolic condensates known as stress granules (related to but distinct from the G3BP-marked stress granules in mammalian cells). In the revised manuscript we show that Pab1-mScarlet forms constitutive cytosolic foci even in glucose-containing media in the 2x*SUP35pr-Sis1* cells. Also, unlike wild type, non-inducible Sis1 cells are unable to increase their stress granules in EtOH/glycerol media. Sis1 has recently been shown to be an essential factor in the biochemical regulation of stress granules (Yoo et al., 2022, Mol Cell). In light of this work, we interpret our results to suggest that proper regulation of Sis1 expression levels by Hsf1 is required for proper regulation of stress granules in environments with changing carbon sources.

3. Additional controls are needed to establish that the 2XSUP35prom-SIS1 strain does not have unlinked respiratory defects that might account for some of the observed phenotypes.

To determine whether the growth phenotypes displayed by the 2x*SUP35pr-SIS1* strain are linked to Sis1 expression, we first examined the parent strain in its lineage as we expected off target mutations would most likely have occurred during the genome editing step in which we generated the *1xSUP35pr-SIS1* strain. We created this strain via Cas9-mediated cleavage of the endogenous *SIS1* promoter followed by homology directed repair with a template containing the *SUP35* promoter. In the original submission, we reported our observation that the 1x*SUP35pr-SIS1* strain had a slow growth phenotype on YPD (glucose-containing) plates and expressed Sis1 to ~0.5x the level of wild type (Figure S2). Thus, we expected the 1x*SUP35pr-SIS1* strain to recapitulate – or even exacerbate – the slow growth phenotype of 2x*SUP35pr-SIS1* in EtOH/glycerol regardless of whether the phenotypes were linked to Sis1 expression. However, we observed the opposite result: despite its slow growth in glucose, 1x*SUP35pr-SIS1* grows indistinguishably from wild type in EtOH/glycerol, completely rescuing the phenotype. Thus, the growth phenotype is not coming from the Cas9-mediated edit at the endogenous locus, but must be coming from the second, ectopic copy of *SUP35pr-SIS1* integrated at the *trp1* locus – either from too much Sis1 expression or an effect of the integration construct. After ruling out integration into the *trp1* locus or restoration of *TRP1* prototrophy, we tested whether 2x*SUP35-SIS1* might express too much Sis1 to grow well in EtOH/glycerol by expressing slightly more *SIS1* in an attempt to exacerbate the phenotype. To this end, we went back to the 1x*SUP35-SIS1* strain and integrated a second copy of *SIS1* in the *trp1* locus as before – but this time under the endogenous *SIS1* promoter to give higher total Sis1 expression. Indeed, this strain failed to grow altogether in EtOH, consistent with too much Sis1 expression underlying the phenotype. These results are now included in Figure S3. Taken together, these data suggest that the fitness-conferring role of regulation of *SIS1* expression by Hsf1 is to coordinate Sis1 expression with a larger metabolic program rather than simply to drive Sis1 expression during acute stress. We have changed the title of the paper to reflect this nuance by replacing “Induction of Sis…” with “Transcriptional regulation of Sis1…”.

Reviewer #1 (Recommendations for the authors):The two main weaknesses are the lack of insight into how Sis1 is acting to promote fitness under heat shock conditions, and whether sequestration of Sis1at the nucleolar periphery is really a viable mechanism for preventing its actin in the nucleus, or whether some other event such as an inactivating post-translational modification of Sis1 under heat shock conditions might be involved. Further experiments to address these issues are recommended.

We hope that the new experiments and analysis have satisfied these recommendations.

Reviewer #2 (Recommendations for the authors):1. The experimental result in Figure 2B shows that Hsf1 activity increases linearly during the entire time course (90 min) following Sis1 depletion. No plateau is reached within this time-frame. Yet, the model in Figure 2C predicts that a plateau should be reached. This impacts on the usefulness and how faithfully the model recapitulates Hsf1 regulation. Explaining the discrepancy between experiment and model would strengthen the paper. (Perhaps Sis1 depletion results in collapse of the proteostasis network and indirect effects due to massive protein misfolding takes over, which makes interpretation of later time points not very meaningful.)

The reviewer is correct: the discrepancy between the simulation and experiment of anchoring away Sis1 at the later timepoints means either that we are modeling the role of Sis1 in basal Hsf1 repression incorrectly or that anchoring away Sis1 has a slower time scale indirect effect on Hsf1 activity – due for instance to proteostasis collapse. We have now mentioned this in the text describing Figure 2B.

2. In Figure 2D experimental variation in Hsf1 activity after heat-shock is documented and is the simulated in Figure 2E by altered rates of protein synthesis in the model. The authors claim that initial metabolic states of the cells are causing the experimental variation by impacting on translation (lines 192-204). I suggest to directly test this hypothesis by simply applying stricter conditions during microbiological culturing. In the Methods section (Lines 600-605) it is stated that cells were cultured at room temperature and that over-night cultures were used. Cells from over-night cultures may have variable metabolic states due to variable nutrient (glucose) consumption and room temperature may or may not be stable adding to the variability. Lines 608-611 describe the use of a mild 1 h pre-heat-shock (30 degree C) in a PCR tube with a relatively large volume of culture (750 microL). This raises the question if cells even get enough oxygen under these conditions? Repeating the experiment with cultures grown to log phase from an overnight culture for at least three generations (OD600=0.1 to = 0.8) at a stable temperature provides a test to the claim that the variability comes from the metabolic state of the cultures, i.e. experimental variation should decrease if the hypothesis is true.

Indeed, careful outgrowth obviates these effects. The point here is that such metabolic variation is easy to capture in the model by modulating the rate of protein synthesis.

3. The authors show that the 2XSUP35pr-SIS1 strain does not undergo diauxic shift and grows poorly on non-fermentable carbon sources (Figure 5A-B, lines 327-346). This finding is consistent with the claim that Sis1 induction by Hsf1 promotes fitness in the presence of non-preferred carbon sources. Yet the SIS1 promoter is also regulated by Msn2 and Msn4 that are activated when cells grow on non-preferred carbon sources. It may be Msn2 and Msn4 and their binding sites that explain the phenotype rather than decoupling from Hsf1 regulation. It would be valuable to distinguish between these possibilities so the study accurately reflects the involvement of Hsf1/HSE in the phenotype.

*SIS1* is an exclusive target of Hsf1 and has no known binding sites for Msn2/4 in its promoter (see Solís et al., Mol Cell 2016, Figure 2B).

4. A concern I have is that the 2XSUP35pr-SIS1 strain has lost respiratory capacity during strain construction and that this phenotype is not genetically linked to the SIS1 promoter replacement. The appearance of respiratory deficient phenotypes is quite common when transforming yeast cells and the strain has undergone several sequential steps of genetic modification, some of which include slow growth. In light of the central part this strain plays in the study I strongly suggest ruling out this scenario by performing additional controls. For example, complementation of the respiratory phenotype with WT SIS1 or alternatively repair of the promoter could be potential routes. Strain crosses and analysis of the meiotic offspring is another way.

Please see our response to essential revision point 3 above.

Recommended text changesa) Lines 26-28. Rephrase "Unexpectedly". I got stuck on here when reading it for the first time. It is probably unclear for most readers why this is unexpected and I actually do not find it to be unexpected.

We have reworded the manuscript.

b) Lines 103. Rephrase "However" since the previous sentence is not contrasted. I got stuck on here when reading it for the first time.

This has been changed.

c) Lines 227-253 contains a lengthy description of the construction and rationale behind the strain referred to as 2xSUP35pr-SIS1. I appreciate all the work behind the strain construction but the description adds little to the Results. I suggest editing out this part and moving any needed parts to the Methods section so that the paragraphs starting at line 216 and 255, respectively, become merged to one single paragraph.

We have moved this description to the methods.

Reviewer #3 (Recommendations for the authors):1. It is unclear how the misfolding rates (Kdup) at different temperatures are calculated and modeled. Clearly, the experimental results appear to mirror the modeling output but how the authors got there is not apparent.

The rate is determined based on an empirically derived function relating temperature and protein misfolding as previously modeled in the original manuscript. This has been clarified in the text.

2. The Hsp70∆FBL strain is not explained in the text and will be confusing to readers unfamiliar with previous work from this group.

We have included a lengthier description of the Hsp70∆FBL strain.

3. The Sis1 localization experiments in Figure 4B appear to show significant nuclear localization rather than exclusively nucleolar peripheral signal. It might be useful to include a DAPI stain or nucleoplasmic tagged protein to conclusively state that Sis1 is nucleolar.

Please see new data in the manuscript and descriptions above.

4. Also, in Figure 4B, I expected to see Ssa1 exhibit nucleoplasmic localization in addition to cytoplasmic puncta. At the resolution provided, it's hard to discern any significant difference between Sis1 at the nuclear/nucleolar level, yet the results are interpreted in opposite directions – Sis1 is not where it needs to be to regulate Hsf1, but Ssa1 is. Is it possible to perform ChIP experiments to discern whether Sis1 is present at HSEs with Hsp70 and Hsf1 during the attenuation phase?

Please see the new colocalization data with Hsf1.

5. As stated above, it is not clear what the reason is for the post-diauxic shift and ethanol/glycerol growth phenotypes. Sis1 is an essential gene, so it is not surprising that perturbations in expression might lead to growth defects under non-ideal conditions. To claim this is an important "fitness" outcome related to Hsf1-mediated control of SIS1 gene expression is an overinterpretation of the data. The fact that the Hsp70∆FBL strain exhibits no such problems argues against a need for precise Hsf1 regulation under these growth conditions.

Please see our responses to essential revisions 2 and 3 above.